# Health and Wellbeing of Regional and Rural Australian Healthcare Workers during the COVID-19 Pandemic: Baseline Cross-Sectional Findings from the Loddon Mallee Healthcare Worker COVID-19 Study—A Prospective Cohort Study

**DOI:** 10.3390/ijerph21050649

**Published:** 2024-05-20

**Authors:** Mark McEvoy, Gabriel Caccaviello, Angela Crombie, Timothy Skinner, Stephen J. Begg, Peter Faulkner, Anne McEvoy, Kevin Masman, Laura Bamforth, Carol Parker, Evan Stanyer, Amanda Collings, Xia Li

**Affiliations:** 1La Trobe Rural Health School, La Trobe University, Bendigo, VIC 3550, Australia; s.begg@latrobe.edu.au; 2Swan Hill District Health Service, Swan Hill, VIC 3585, Australia; gcaccaviello@shdh.org.au; 3Bendigo Health, Bendigo, VIC 3550, Australia; acrombie@bendigohealth.org.au (A.C.); pfaulkner@bendigohealth.org.au (P.F.); kmasman@bendigohealth.org.au (K.M.); lbamforth@bendigohealth.org.au (L.B.); cparker@bendigohealth.org.au (C.P.); estanyer@bendigohealth.org.au (E.S.); acollings@bendigohealth.org.au (A.C.); 4School of Psychology and Public Health, La Trobe University, Bendigo, VIC 3550, Australia; t.skinner@latrobe.edu.au; 5Kyabram District Health Service, Kyabram, VIC 3620, Australia; amcevoy@kyhealth.org.au; 6Department of Mathematics and Statistics, La Trobe University, Melbourne, VIC 3550, Australia; x.li2@latrobe.edu.au

**Keywords:** healthcare workers, psychological distress, wellbeing, COVID-19

## Abstract

Background: Coronavirus 19 (COVID-19) has created complex pressures and challenges for healthcare systems worldwide; however, little is known about the impacts COVID-19 has had on regional/rural healthcare workers. The Loddon Mallee Healthcare Worker COVID-19 Study (LMHCWCS) cohort was established to explore and describe the immediate and long-term impacts of the COVID-19 pandemic on regional and rural healthcare workers. Methods: Eligible healthcare workers employed within 23 different healthcare organisations located in the Loddon Mallee region of Victoria, Australia, were included. In this cohort study, a total of 1313 participants were recruited from November 2020–May 2021. Symptoms of depression, anxiety, post-traumatic stress, and burnout were measured using the Patient Health Questionnaire-9 (PHQ-9), Generalised Anxiety Disorder-7 (GAD-7), Impact of Events Scale-6 (IES-6), and Copenhagen Burnout Inventory (CBI), respectively. Resilience and optimism were measured using the Brief Resilience Scale and Life Orientation Test—Revised (LOT-R), respectively. Subjective fear of COVID-19 was measured using the Fear of COVID-19 Scale. Results: These cross-sectional baseline findings demonstrate that regional/rural healthcare workers were experiencing moderate/severe depressive symptoms (n = 211, 16.1%), moderate to severe anxiety symptoms (n = 193, 14.7%), and high personal or patient/client burnout with median total scores of 46.4 (IQR = 28.6) and 25.0 (IQR = 29.2), respectively. There was a moderate degree of COVID-19-related fear. However, most participants demonstrated a normal/high degree of resilience (n = 854, 65.0%). Based on self-reporting, 15.4% had a BMI from 18.5 to 24.9 kgm^2^ and 37.0% have a BMI of 25 kgm^2^ or over. Overall, 7.3% of participants reported they were current smokers and 20.6% reported alcohol consumption that is considered moderate/high-risk drinking. Only 21.2% of the sample reported consuming four or more serves of vegetables daily and 37.8% reported consuming two or more serves of fruit daily. There were 48.0% the sample who reported having poor sleep quality measured using the Pittsburgh Sleep Quality Index (PSQI). Conclusion: Regional/rural healthcare workers in Victoria, Australia, were experiencing a moderate to high degree of psychological distress during the early stages of the pandemic. However, most participants demonstrated a normal/high degree of resilience. Findings will be used to inform policy options to support healthcare workers in responding to future pandemics.

## 1. Background

The highly contagious novel coronavirus 19 (COVID-19) has created extraordinary pressure and highly complex challenges for healthcare across the globe [1]. As the outbreak swiftly became a global pandemic, the narrative surrounding COVID-19 continued to evolve—shaping both the healthcare system and now the public’s ‘COVID normal’. However, throughout this evolution the research community has lagged in understanding how the health, wellbeing, and resilience of healthcare workers has been affected in the context of rurality and remoteness during a pandemic.

From the onset of the pandemic, healthcare workers have been placed at the frontline, bearing the brunt of the virus’ demands. Australian healthcare workers faced this pandemic on the back of some of the most devastating sequences of climatic disasters including drought, bushfire, and flooding [2]. Of the Australian communities affected by the consequences of these passing events, those people living in regional, rural, or remote locations are more exposed to the impacts of these events than their metropolitan counterparts.

According to the Australian Institute of Health and Welfare’s report on ‘Rural and Remote Health’ (2020) [3], 28% of Australians live in either rural or remote areas across multiple, diverse populations. These communities, even without the threat of COVID-19, are regularly faced with unique healthcare challenges and poorer health outcomes [3]. Rural and remote healthcare organisations act as the community’s largest employment provider [4], highlighting the major potential for significant impacts on the regional/rural/remote healthcare workforce when a pandemic of this scale erupts. In addition, there are limited published studies in the current literature that examine the immediate and long-term effects of COVID-19 on healthcare workers in these geographical locations. One longitudinal, mixed-methods study of Tasmania nurses by Marsden et al. (2022) described the stressors and manifestations of them in the context of home, work, and the social environment [5]. Although this was only during the first year of pandemic, it does provide a basis for comparison, as Tasmania, according to the Australian Bureau of Statistics, has a population with a similar geographic spread to the Loddon Mallee in terms of rurality.

Most published research evaluating the effects of COVID-19 on healthcare workers is cross-sectional and has focused exclusively on metropolitan populations. Additionally, study samples mainly include specialised physicians and intensive care unit (ICU) nursing staff and exclude other groups such as allied health and non-clinical healthcare workers. This arguably creates a misrepresentation of the healthcare workforce and also omits a diverse group of staff who are essential to delivering an effective health service [6]. However, there are exceptions with some longitudinal studies that include allied health and other support staff who work in the health system in their cohort, Lamb et al. (2022) being one such study [7]. Whilst this study is set in England, which had a very different pandemic experience to Australia, it does provide an indication of the impact of the pandemic across the health system as a whole and has a large sample to support their findings. This study highlights the need for support and clear guidelines to help staff navigate times of distress and suggests that this needs to be a part of planning for the future. McGuiness et al. (2023) is a longitudinal cohort study undertaken in Metropolitan Melbourne, Victoria, that looked more broadly at the health sector, and whilst they had 1667 unique responses, only 496 completed all three surveys [8]. Apart from the metropolitan context, a further confounding factor in using this study as a comparison is the number of lockdowns experienced in Melbourne compared to regional Victoria, with fewer lockdowns experienced in regional Victoria. The LMHCWCS fills an important gap in our current understanding of the effects of the COVID-19 pandemic by being conducted exclusively in a regional/rural setting and including both clinical and non-clinical healthcare workers. It also provides an opportunity for significant longitudinal research on the long-term effects of the pandemic on a uniquely regional and rural healthcare worker sample; research that to date has been scarcely reported in Australia [4]. The LMHCWCS is collecting information from healthcare workers employed within the Loddon Mallee region, Victoria, Australia.

The COVID-19 pandemic has placed more pressure on the Loddon Mallee region healthcare workforce than ever before and rural and regional health services are working through a backlog of treatment and care delayed by the pandemic [9]. It is anticipated that this cohort will provide an understanding of the protective and resilience factors that result in better health and wellbeing outcomes for healthcare workers within these regional and rural communities in the face of the mass disruption associated with the COVID-19 pandemic. In turn, this will help create evidence-based recommendations for policies and practice to better support regional and rural health workers now and when faced with other pandemics into the future [4].

## 2. Method

### 2.1. Aims

The aims of the current study are as follows:

To cross-sectionally describe the health and wellbeing of the rural/regional healthcare workforce during the early stages of the COVID-19 pandemic using baseline data from the LMHCWCS cohort.

### 2.2. Design and Setting

The LMHCWCS is a prospective longitudinal cohort study exploring the immediate and long-term impacts of the COVID-19 pandemic on the health and wellbeing of regional and rural healthcare workers. The findings for this current investigation pertain to cross-sectional data from a baseline survey conducted as part of the broader longitudinal study. Healthcare workers currently employed or volunteering within the Loddon Mallee region of Victoria, Australia, were recruited from 23 health organisations, including 17 regional and rural health services (16 public hospitals and 1 non-public hospital), 5 community health centres, and the Murray Primary Health Network (MPHN). Full details of the study design and methods are available in the study protocol [4].

Data were collected through self-administered online questionnaires at baseline (November 2020–May 2021). Previously validated and reliable instruments were used to measure various health, wellbeing, and resilience outcomes for participants during the COVID-19 pandemic.

### 2.3. Participant Eligibility

Eligible participants during the baseline recruitment period needed to provide informed consent, be aged 18 or over, have the ability to access and use an electronic device, and be currently employed in any capacity (clinical or non-clinical) or volunteering within a participating Loddon Mallee health service [4]. As it is a condition of employment that healthcare workers must be able to read/speak English and be able to use an electronic device (i.e., computer, mobile phone), all potential participants were eligible and hence no healthcare workers had to be actively excluded on this basis.

### 2.4. Recruitment

The sampling frame consisted of a workforce of 8107 healthcare workers in public hospitals in the Loddon Mallee region and an unknown number of healthcare workers working within non-public hospitals, community health organisations, and primary healthcare providers in the region. All eligible healthcare workers at each of the public hospitals and an unknown number of healthcare workers within the non-public hospitals, community health organisations, and primary healthcare providers were invited to participate in the study. An online link directed potential participants to a consent form and questionnaire sent via staff email systems.

Baseline recruitment commenced on 30 November 2020 and ceased on 31 May 2021. Within this time, Victorians experienced two state-wide COVID-19 lockdowns [10]. A timeline of pandemic-related events illustrating the events that occurred during the baseline data collection period is summarised in Figure 1 as a means of providing a greater understanding the specific challenges faced by regional and rural healthcare workers during the pandemic. Pre-notification and reminder emails, as well as a random draw incentive, were employed to encourage participation in the study.

Study participants were recruited through three different approaches: (1) online invitations sent through the health organisation’s direct staff emails; (2) a multimodal recruitment media campaign, including Trialfacts online recruitment targeting participating health service employees [11]; and (3) face-to-face recruitment presentations (see Figure 2).

All three recruitment approaches involved the incentive of winning a AUD 1000 grocery voucher for enrolling in the study. Where a member of the study team was present to promote the study to potential participants, for ethical reasons, no recruitment was made directly through this contact. Rather directions were given on how to access and complete the online questionnaire previously sent to them through staff email and other means.

### 2.5. Study Sample

Email invitations were sent to all employees, (n = 8107) in public hospitals in the Loddon Mallee region. An unknown number of email invitations were also extended to non-public hospitals, community health organisations, and primary healthcare providers in the region through the staff email system and staff newsletters. A total of 2735 potential participants clicked on the baseline questionnaire link: 1174 from public hospitals (14.5% of this population), 286 from non-public organisations, and 1275 from other organisations. The number who completed the consent was 1043 from public hospitals (12.9% of this population); 266 from non-public hospitals, community health organisations, or primary healthcare providers; and 4 who were missing employer information. Of the 1313 consenting participants, 97.1% provided usable complete data.

### 2.6. Sample Weighting

The sample consists of respondents from public hospitals (79.4%) and non-public hospitals, community health organisations, and primary healthcare providers (20.3%). Given that there were Victorian state government workplace characteristics data on the sampling frame for the public hospital participants, and that these people represent the majority of the sample, the public hospital participant part of the sample was weighted for non-response using these data. Weighting for the non-public hospitals, community health organisations, and primary healthcare provider part of the sample was not performed as no workplace data were available.

Public hospital sample weights were calculated by raking, using four known demographic factors in the public hospital population from the annual Victoria’s Health and Human Services Knowledge Bank, Information Portal, as of July 2020 [12]. The inverse probability weighting of the data was chosen instead of the multiple imputations of missing data as a means of assessing the potential effects of sampling bias and has been shown to be comparable to multiple imputations for assessing the likely impacts of missing data [13]. The four-dimensional raking residuals were calculated using age, sex-at-birth, profession, and hospital size. These variables were available for 80.5% (n = 6528) of the sampling frame. Matching public hospital participant sample weights were calculated with SPSS 28.0.0 [14] and R 4.1.2 [15].

### 2.7. Follow-Up Assessments

Self-administered online questionaries will be used for the follow-up of the cohort at 6, 12, and 24 months after baseline recruitment. The same/similar questionnaires (including the same psychometric scales) as used in the baseline data collection will be readministered to study participants at each follow-up timepoint. Each participant will receive their follow-up questionnaire at 6, 12, and 24 months from the time they completed their baseline questionnaire. Reminder strategies and continued participant updates through social media and participant newsletters are being used to encourage continued participation at critical time points and to reduce attrition.

### 2.8. Study Outcome and Outcome Measurement Instruments

To date there is no ‘core outcome set’ for examining the health and wellbeing of healthcare workers [4]. As such, an outcome taxonomy was created by the study team based on the WHO International Classification of Functioning, Disability, and Health (ICF) [16] and Williamson and Clarke taxonomies [17] and constructed according to standards developed by the Core Outcomes Measures in Effectiveness Trials (COMET) initiative [17,18].

### 2.9. Study Outcomes

The complete list of study outcomes has been described elsewhere [4]. For the current investigation, the primary study outcomes are grouped according to (1) emotional health and wellbeing; and (2) lifestyle and physiological outcomes.

#### 2.9.1. Emotional Health and Wellbeing Outcomes

Emotional health and wellbeing were measured using self-reported validated measures of wellbeing, generalised anxiety, fear of COVID-19, post-traumatic stress symptoms (PTS), depressive symptoms, burnout, psychological stress, resilience, mental fatigue, isolation and loneliness, and optimism.

#### 2.9.2. Lifestyle and Physiological Outcomes

Lifestyle and physiological outcomes include body mass index, alcohol intake, smoking, dietary intake, and sleep quality.

### 2.10. Outcome Measuring Instruments and Data Analysis

Subjective wellbeing was measured using the single life satisfaction question from the Personal Wellbeing Index—Adult (Cronbach’s alpha (α) = 0.80) [19]. This question asks participants to rate their level of life satisfaction with the following question “How satisfied are you with your life as a whole?” Response options range from 0–10 with zero indicating no satisfaction and 10 indicating complete satisfaction, i.e., higher scores indicate greater wellbeing. Depressive symptoms were measured using the Patient Health Questionnaire—9 (PHQ-9) (α = 0.89) [20], range 0–27, with higher scores indicating greater degree of depressive symptoms; score ≥ 10 indicating moderate-to-severe depressive symptoms. Anxiety symptoms were measured using the Generalised Anxiety Disorder—7 (GAD-7) (α = 0.92) [21], range 0–21, with higher scores indicating greater degree of anxiety symptoms and a score ≥ 10 indicating moderate-to-severe anxiety symptoms. Post-traumatic stress was measured using the Impact of Events Scale—6 (IES-6) (α = 0.80) [22], range 0–24, with higher scores indicating greater degree of post-traumatic stress and a score >9 indicating moderate-to-severe post-traumatic stress (PTS) reactions. Personal (α = 0.87) and patient/client (α = 0.74) burnout was measured using the Copenhagen Burnout Inventory (CBI) [23], range 0–100, with higher scores indicating greater burnout. Resilience was measured using the Brief Resilience Scale (α = 0.80) [24], range 1–5, with higher scores indicating greater psychological resilience; categorised as low (scores 1.00–2.99), normal (scores 3.00–4.30), and high resilience (4.31–5.00), respectively. Optimism was measured using the Life Orientation Test—Revised (LOT-R) (α = 0.78) [25], range 0–24, with higher scores indicating higher optimism; categorised as low (score < 14), moderate (scores 14–18), and high optimism (Score > 18), respectively. Subjective fear of COVID-19 was measured using the Fear of COVID-19 Scale (α = 0.82) [26], range 7–35, with higher scores indicating greater COVID-19-related fear. Fatigue was measured using the vitality question of the Short Form—12 (SF-12) (α = 0.89) [27]. This question asks participants to rate their energy levels in the last 4 weeks using the following question “How much of the time during the past 4 weeks did you have a lot of energy?” Response options were “all of the time”, “most of the time”, “a good bit of the time”, “some of the time”, “a little of the time”, and “none of the time”. Isolation and loneliness were measured using the UCLA loneliness score version 3 (α = 0.72) [28], range 3–9, with higher scores indicating greater isolation and loneliness.

A comparison of emotional health and wellbeing outcomes between four different occupational groups (medicine, nursing, allied health, and non-clinical) was made. Regression models adjusted for potential confounders (i.e., age, gender, modified Monash category, education) were used to examine differences. Age, gender, modified Monash category, and education were included as categorical predictors in each model. For continuous outcomes, linear mixed models (lmer function in R lme4 package) were developed, including random intercepts for healthcare site (20 healthcare sites and 4 individuals have missing healthcare site). Results are expressed as adjusted differences in mean outcomes and 95% confidence intervals between occupational groups, using nursing (largest sample size) as the reference category. A Type III Analysis of Variance with Satterthwaite’s method was used for fixed-effect terms tests, including occupation variables. For binary outcomes, random effects models were developed (glmer function in R) using binomial distribution with logit link to estimate adjusted risk ratios and 95% confidence intervals. Type III Wald chi-square tests were used to obtain fixed-effect terms tests.

Self-reported anthropometric and lifestyle behaviours included body mass index (BMI), dietary consumption, smoking tobacco, alcohol consumption, and sleep quality. Body mass index was calculated from self-reported height and weight and categorised according to World Health Organisation (WHO)-recommended criteria [29]. Dietary intake was measured using the Australian short diet questions [30]; fruit and vegetable consumption was measured using the following questions: “How many serves of vegetables do you usually eat each day? (a ‘serve’ = 1/2 cup cooked vegetables or 1 cup of salad vegetables)” and “How many serves of fruit do you usually eat each day? (a ‘serve’ = 1 medium piece or two small pieces of fruit or 1 cup of diced pieces)”. Response options for each question were “1 serve or less”, “2–3 serves”, “4–5 serves”, “6 serves or more”, and “Do not eat vegetables/fruit”. Tobacco intake was measured using the American Cancer Society smoking questionnaire [31] and categorised as past, current, and never smoked. Alcohol intake was measured using the three-item AUDIT questionnaire [32], range 0–12, with higher scores indicating more risky drinking behaviour; categorised as non-drinker/low-risk (0–4), moderate-risk (5–7), and high-risk (8–12). Sleep quality was measured using the Pittsburgh Sleep Quality Index (PSQI) (α = 0.83) [33], range 0–21, with higher scores indicating poorer sleep quality. Sleep quality was categorised as poor sleep quality and good sleep quality based on the following question for the instrument: “During the past month, how would you rate your sleep quality overall?”

### 2.11. Participant Sociodemographic Characteristics

The baseline sociodemographic characteristics of study participants are described in Table 1. Characteristics and findings are reported for participants working within public hospitals and those working in non-public hospitals, community health organisations, or primary healthcare providers separately due to the weighting applied to the public healthcare part of the sample, and overall sample sociodemographic and occupational characteristics are reported for the total unweighted sample. Sociodemographic characteristics are also compared between four different occupations (medicine, nursing, allied health, and non-clinical) (Table 2). Overall, 80.5% (n = 1057) of the participants are female, with nursing employing the largest proportion of female workers (88.6%). Participants range in age from 18–79 years, mean age of 44.9 years (SD. 12.8), with medicine employing the largest proportion of workers under 45 years of age (61.0%). Most participants live in a MM 2 location (n = 566, 43.1%) with non-clinical workers having the largest proportion of workers living in an MM 3–7 location (53.1%). Approximately 75.0% (n = 988) live in their own home. Less than 1.0% (n = 8) identify as being of Aboriginal and/or Torres Strait Islander origin. Almost 73.0% (n = 957) reported completing a university degree and 2.4% (n = 32) did not complete secondary school level of education. Non-clinical workers were the smallest proportion of workers without a university/other tertiary education (56.3%). Close to 67.0% (n = 884) of participants were either married or in a de facto relationship, 15.9% (n = 209) reported a gross annual income of less than AUD 52,000 per year, while 21.6% (n = 284) reported that they had concerns about their income. Allied health workers were the largest proportion of workers with concerns about their income (25.1%). Home schooling responsibilities were reported by 27.6% (n = 362) of participants, and 28.6% (n = 376) reported carer responsibilities. These results were similar across occupational groups.

### 2.12. Occupational and Workplace-Related Characteristics

The occupational and workplace-related characteristics of study participants and between four different occupations (medicine, nursing, allied health, and non-clinical) are described in Table 2 and Table 3. Most participants were nurses (n = 499, 38.0%), followed by non-clinical staff (n = 435, 33.1%), allied health staff (n = 219, 16.7%), and then medical staff (n = 59, 4.5%). Participants were in part-time (n = 649, 49.4%), full-time (n = 436, 33.2%), casual (n = 72, 5.5%), volunteer (n = 32, 2.4%), retired (n = 11, 0.8%), or other (n = 12, 0.9%) employment. Only 11.7% (n = 153) of participants reported taking direct care of COVID-19 patients with those employed in medicine (54.2%) and nursing (43.3%) having the largest proportion of workers in direct contact with COVID-19 patients.

Half the sample (n = 668, 50.9%) reported that their workplace was very or moderately prepared for the COVID-19 pandemic, while 7.6% (n = 100) reported that their workplace was not prepared at all. There was 64.0% (N = 840) of the sample that were very or moderately confident that the training they received would protect them during the pandemic. Allied health workers had the greatest confidence in infection control training with 72.1% (N = 158) of workers reporting that they were very or moderately confident that the training they received would protect them during the pandemic. Most participants (n = 780, 59.4%) reported that they often or very often had access to the right amount of personal protective equipment (PPE) with 7.1% (n = 92) reporting that they rarely or never had access to PPE. Most had confidence in the PPE training they received (n = 945, 72.0%) and were moderately or very confident in the infection control training (n = 840, 63.7%) that they received. Interestingly, a little more than half of participants (n = 677, 51.6%) were unsure if their workplace had a policy for breaks for those working in full PPE, 7.1% (n = 93), indicated that there was no policy, while almost three-quarters (n = 945, 72.0%) of participants felt that their concerns about PPE were supported by their workplace. Sixteen percent (n = 210) of participants reported that they had been deployed to a new area of work at the start of the pandemic; however, the majority of these participants were confident or very confident in their new role (n = 130, 61.9%). Nursing and allied health workers were more likely to be redeployed. Importantly, most participants did not intend to change their career due to the COVID-19 pandemic (n = 1013, 77.2%). This was similar across occupational groups.

## 3. Results

### 3.1. Emotional Health and Wellbeing

The emotional health and wellbeing attributes of study participants are described in Table 4 and Table 5. Symptoms of psychological distress measured by objective and validated scales indicated that participants were experiencing moderate to severe depressive symptoms (n = 211, 16.1%), moderate to severe anxiety symptoms (n = 193, 14.7%), and high personal or patient/client burnout with median total scores of 46.4 (IQR = 28.6) and 25.0 (IQR = 29.2), respectively. Those employed in nursing had the largest proportion of workers experiencing moderate/severe symptoms of depression (21.0%) and anxiety (19.0%), while medical workers reported the largest level of personal or patient/client burnout with median total scores of 64.3 (IQR 16.0) and 29.2 (IQR 26.6), respectively (Table 5). Many participants also reported feeling fatigued a good bit of the time, most of the time, or all the time (n = 543, 41.4%). This was similar across occupational groups. Surprisingly, 17.5% (n = 230) participants were already experiencing symptoms associated with moderate-to-severe post-traumatic stress (PTS) and this was similar across occupational groups. Despite the relatively high level of symptoms of psychological distress, most participants demonstrated a normal or high degree of resilience (n = 854, 65.0%). Workers employed in medicine had the largest proportion of workers with normal/high resilience (90.0%), while allied health workers had the lowest (78.0%). However, only 20.1% (n = 264) of participants reported a moderate or high degree of optimism, with those employed in medicine reporting the lowest (10.2%). Participants had a high degree of personal wellbeing with a mean life satisfaction score of 7.2 (SD. 1.8) and a range of 0–10. Those employed in medicine had the highest mean level of life satisfaction of 7.7 (SD. 1.4), while those in nursing experienced the lowest with a mean score of 7.2 (SD. 1.9). Despite this, many participants reported feeling isolation or loneliness with a median total score of 5.0 (IQR = 3.0) and a range of 3–9. This was similar across occupational groups. Participants also reported a moderate degree of COVID-19-related fear with a mean of 13.1 (SD. 5.1) and a range of 7–34. Allied health workers experienced the highest level of fear with a mean score of 13.6 (SD. 5.3), while those in medicine experienced the lowest with a mean score of 11.2 (SD. 4.0). All psychometric instrument scales had acceptable internal consistency (α > 0.70).

Occupational group differences in mean scores and the proportions of emotional health and wellbeing outcomes are described in Table 6. Statistically significant adjusted differences between groups were only observed for life satisfaction, fear of COVID-19, and optimism. The largest difference in life satisfaction was observed between medicine and nursing workers. The largest differences in fear and optimism were observed between medicine and nursing workers and medicine and allied health workers. Important, but not statistically significant, differences were also observed for burnout, with those working in medicine generally experiencing higher levels of burnout but, surprisingly, higher levels of life satisfaction.

The comparison of HCWs working within the private hospital or community health services settings with those working within the public hospital setting indicates that those in private hospital or community health services were marginally older, allied health or non-clinical workers, in full-time employment, and were less likely to have access to PPE, less confident in infection control training, and less likely to have direct care of COVID-19 patients. These HCWs were less likely to be experiencing moderate/severe symptoms of depression, anxiety, fatigue, or burnout and were less likely to be experiencing poor sleep but had higher levels of psychological resilience. However, these workers were also more likely to be current smokers and report high-risk alcohol drinking behaviour.

### 3.2. Lifestyle and Physiological Characteristics

The lifestyle behaviours of healthcare workers are described in Table 7. Based on self-reported height and weight, 0.8% (n = 10) of participants have a body mass index (BMI) of <18.5 kgm^2^, 15.4% (n = 202) have a BMI from 18.5 to 24.9 kgm^2^, and 37.0% (n = 486) have a BMI of 25 kgm^2^ or over, indicating overweight or obesity. Overall, 7.3% (n = 96) of participants reported they are current smokers and 20.6% (n = 271) reported alcohol consumption that is considered moderate- to high-risk drinking. Only 21.2% (n = 278) of the sample reported consuming four or more serves of vegetables daily and 37.7% (n = 496) reported consuming two or more serves of fruit daily. Almost half the sample reported having poor sleep quality (n = 631, 48.0%).

## 4. Discussion

There are several studies describing the impacts of the COVID-19 pandemic on the mental health and wellbeing of healthcare workers within different professional groups and settings from within Australia [34,35,36,37,38] and many more internationally [39]. This study provides a unique perspective into the impacts of the pandemic on over 1300 clinical and non-clinical healthcare workers working within a regional and rural setting during the early stages of the pandemic in Australia. Findings from within a regional or rural setting have been scarcely reported in the literature, either in Australia or internationally. Furthermore, this study offers comprehensive findings on health and wellbeing, including lifestyle behaviours, which are seldom collected and reported together.

The proportions of moderate to severe degrees of anxiety symptoms, depressive symptoms, symptoms of PTS, and burnout within regional and rural healthcare workers during the COVID-19 pandemic in Victoria are concerning; however, the proportions of workers with these symptoms were generally lower that those reported by other Australian studies [35]. Furthermore, the higher levels of anxiety and depressive symptoms experienced by nursing workers are in line with findings from other studies [39]. The higher levels of personal or patient/client burnout in medical workers has also been commonly reported [40]. These levels of burnout in medical staff might be due to staff shortages in regional/rural Australia, with these staff having longer travel times to work, greater workloads, longer working hours, and additional responsibilities compared with their metropolitan counterparts [41].

These differences between studies are likely attributable to different data collection times during the pandemic, measurement tools, and population structures, with the current study consisting of approximately one-third non-clinical healthcare workers who may not be at high risk of occupational COVID-19 exposure. Most participants were recruited and completed their questionnaires between the second and third public health state lockdowns at a time when healthcare workers may have been experiencing less concerns about COVID-19 infection (Figure 1). However, the third and fourth public health state lockdowns occurred in February and May 2021, respectively, with a noticeable decline in recruitment numbers during this timeperiod. Victoria’s vaccine rollout in February 2021, during the recruitment period, may have also contributed to the lower levels of psychological distress reported in this study. Finally, although Victoria experienced significant bushfire and flooding events in 2019–20 and 2021, the Loddon Mallee region was not directly impacted. Moreover, none of these natural disasters occurred during the data collection period. Therefore, it is unlikely that the study findings have been impacted by these events.

The lower prevalence of distress and lower burnout scores in healthcare workers within the private hospital and community health settings compared with public hospital healthcare workers may be due to a lower perceived risk of a COVID-19 infection as COVID-19 patients only initially presented to public hospitals during the pandemic in Australia. These findings might also be influenced by different working conditions such as reduced overall caseloads and higher quality working environment due to fewer working hours and better working conditions [42,43].

Workplace pandemic preparedness and the availability of appropriate PPE are also associated with lower degrees of psychological distress [44], with almost 70% healthcare workers in this study reporting that their workplace was somewhat to very prepared and less than 20% of workers reporting an insufficient supply of PPE. Furthermore, the risk of infection in this study was lower than that reported in large tertiary hospitals in the state capital, Melbourne [34], as only 11.7% of healthcare workers reported taking direct care of COVID-19 patients.

A key finding of this study is that 65.0% of healthcare workers considered themselves to have a normal or high level of psychological resilience, with medical workers having the greatest level of resilience. This suggests that the relatively high level of resilience within the study population may be an additional reason for the lower degree of psychological distress compared with other studies. This relatively high level of psychological resilience has also been reported in other Australian studies [34,35,36,37,45]; however, it did not always correspond with low levels of psychological distress [34]. By comparison, a recent meta-review of 40 systematic reviews (1828 primary studies) of the mental health of healthcare workers globally, during the COVID-19 pandemic, reported a pooled prevalence for anxiety, depression, stress/post-traumatic stress symptoms, and burnout of 16–41%, 14–37%, 18.6–56.5%, and 12–36%, respectively [39]. The prevalence of the corresponding measures of psychological distress in the current study are at the lower end of the prevalence range reported in this meta-review.

Social support is an important component of overall wellbeing and a lack of social support and isolation/loneliness are well-known risk factors for psychological distress [46] [47]. The moderate to high degree of loneliness observed in the current study during the COVID-19 pandemic has also previously been reported in Australia [35] and internationally [48,49]. This may have been the result of the severe public health restrictions experienced by residents of Victoria during the pandemic, as well as an increased need to quarantine from family and friends to prevent the transmission of the virus. Given the impact that loneliness can have on the wellbeing of healthcare workers, interventions, such as peer-support programs, aimed at improving social support (especially during a pandemic) may be helpful for reducing the degree of psychological distress and improve the quality of life of healthcare workers during times of crises.

Of note is the relatively low level of optimism reported by healthcare workers in this study with only one-fifth of participants reporting high feelings of optimism about the future. Medical workers were the least optimistic. This is surprising considering that this occupation group had the highest level of resilience and life satisfaction. By comparison, a higher proportion of optimism was reported by another Australian study conducted within Melbourne metropolitan healthcare workers around the same time as the current study, with one-third of healthcare workers experiencing high feelings of optimism about the future [35]. However, in the study by McGuiness et al. (2021), those working in medicine reported higher levels of optimism than nursing and allied health workers. The differences between studies in the proportion of healthcare workers experiencing high feelings of optimism, especially in medical workers, are likely to be attributable to different measuring instruments, with McGuinness, S., et al. (2021) [35] using a visual analogue scale to measure optimism about the future. However, the low level of optimism reported by medical workers in this study may also be due to the corresponding high burnout observed, possibly due to increased workload and demands due to under resourcing in regional/rural areas.

Moderate to high degrees of COVID-19-related fear have been reported across the world during the pandemic [50]. In Australia, several studies have measured and reported COVID-19-related fear in healthcare workers [45,51,52,53,54]. All studies, except one [54], included healthcare workers and the general public/patients in the study population, with mean COVID-19-related fear scores, measured with the Fear of COVID-19 Scale, ranging from 15.0 to 18.4. The total Fear of COVID-19 Score of 13.1 observed in the current study is less than that reported by previous studies and may be one explanation for the lower prevalence of psychological distress observed in this study. The lower prevalence of COVID-19-related fear may be due to the regional/rural locations of the health services where COVID-19 cases were lower and the high level of health service preparedness perceived by healthcare workers. The relatively lower level of COVID-19-related fear was also observed in healthcare workers residing in south–west Victoria around the same time as the current study [45]. The differences in the fear of COVID-19 scores observed between medicine and nursing workers and medicine and allied health workers observed in this study has been reported in previous studies [55,56]; however, only a few studies used the validated Fear of COVID-19 Scale used in this study.

There are scarce data on lifestyle and physiological characteristics of Australian healthcare workers during the COVID-19 pandemic. Overweight and obesity are highly prevalent in the Australian general population with a prevalence of overweight and obesity of 36.0% and 31.0%, respectively, in 2017–18 [57]. To the best of the author’s knowledge, until now, the prevalence of overweight and obesity in Australian healthcare workers during the COVID-19 pandemic has not been reported. However, a 2020 nationally represented study of Australian workers across multiple industries (including healthcare) reported that 59.0% of Australian employees were either overweight or obese [58]. The proportion of overweight or obesity of 37.0% observed in the current study is lower than the Australian general population and that of Australian workers in general. This is also lower than that reported by international studies [59], except for studies conducted in some Asian populations, where the prevalence of overweight or obesity is quite low [60]. One explanation for the lower prevalence of overweight or obesity in the current study is that BMI was a self-reported measure. It is well known that individuals under-report weight status [61], and this is unlikely to differ among healthcare workers. Another explanation is that regional/rural healthcare workers might have greater access to outdoor recreational activities such as hiking, gardening, or other physical pursuits due to the natural surroundings of rural areas.

Several studies in Australia and internationally have reported an increase in alcohol consumption or high prevalence of alcohol use disorder during the COVID-19 pandemic [62,63,64,65]. This is important to examine, as alcohol use is associated with psychological distress and poor personal relationships [66,67]. Smallwood, N., et al. (2021) reported that 26.3% of Australian healthcare workers increased alcohol use during the second wave of the COVID-19 pandemic, and this was associated with a history of poor mental health and worse personal relationships [62]. The proportion of moderate or high-risk drinking behaviour observed in this study of 20.6% is difficult to compare with the study by Smallwood, N., et al. (2021) as they asked participants whether they increased alcohol consumption or not, and the current study measured this using the AUDIT-3, a validated instrument for quantifying alcohol use and the risk associated with this. However, the proportion of moderate- or high-risk alcohol consumption observed in this study is lower than that reported by studies conducted in the USA around the same time, with Hennein, R., et al. (2020) reporting that 42.6% of health workers had probable alcohol use disorder [63]. The lower proportion of moderate alcohol consumption observed during the pandemic could be due to sampling bias in which more healthy people agreed to complete the study questionnaires or the result of under reporting due to social desirability bias. It might also be the result of limited access to alcohol due to the severe public health restrictions imposed in Victoria or due to greater emphasis on preventive health behaviours because of limited access to services in regional/rural areas. The longitudinal data collection of the current study will examine whether there was an increase in alcohol use over the first two years of the COVID-19 pandemic and, if so, the factors associated with this.

The prevalence of tobacco smoking in healthcare workers during the pandemic has been reported by several studies. In Australia, Stubbs, J.M., et al. (2021) reported that 11.1% of metropolitan hospital healthcare workers were smokers [68], while Rahman, M., et al. (2020) reported a prevalence of smoking of 13.4% [52]. However, the latter study included patients, frontline health and other essential service workers, and community members. Marsden et al. (2022) reported a current smoking prevalence of 8.1% in a sample of public hospital nurses and midwives in the southern region of Tasmania, Australia, in April 2020 [5]. The proportion of tobacco smokers observed in our study of regional/rural healthcare workers of 7.3% is slightly lower than some of the Australian studies but similar to that in Marsden et al. (2022) and may be related to the lower degree of psychological distress observed in this study. Although smoking prevalence is generally higher in adults in outer rural and remote communities, the relatively low prevalence observed in this study might also be due to the healthcare worker study population who may be more aware of the risk of tobacco use and to the low proportion of healthcare worker study participants residing in outer rural and remote communities, with only 17.5% of the study population designated as residing in modified Monash 5–7 categories. Finally, as suggested for alcohol use, the low proportion of current smokers observed in this study might also be due to sampling bias and/or social desirability bias.

To the best of the author’s knowledge, there are no Australian studies that have reported on the eating behaviours of Australian healthcare workers during the COVID-19 pandemic. The low proportion of adequate vegetable and fruit consumption observed in this study is concerning given the growing body of research linking high diet quality with better mental health [69]. These current findings are similar to that reported in the Australian general population in 2020–21, in which only around 6.0% of Australians consume the recommended daily intakes of fruits and vegetables [70]. The low proportion of adequate vegetable and fruit consumption observed is in contrast with the other ‘healthy’ lifestyle behaviours observed in this study and might be the result of demanding schedules with long hours and irregular shifts, leaving many healthcare workers with limited time to prepare or consume healthy meals.

The high proportion of healthcare workers experiencing poor sleep quality (i.e., 48.0%) observed in the current study is similar to that reported by other Australian [68] and international studies [71,72,73,74]. This is also similar to the pooled prevalence of sleep disturbance of 43% (95% CI 36–50) reported by a recent meta-analysis of 18 studies involving nurses [75]. Although stressful events and psychological distress are known to be associated with sleep disturbance [76,77], the high proportion of healthcare workers experiencing poor sleep quality observed in the current study is somewhat surprising given the relatively lower degree of psychological distress in this study population. Previous studies have reported that factors associated with reduced sleep quality included changes in sleeping habits, shift work, anxiety, fear driven by COVID-19 news and lack of treatment knowledge, female gender, monthly income, isolation [78], perceived income, hours spent outdoors, the number of family/friends with COVID-19, and history of depression [79]. Further research is needed to determine which factors have had the greatest impact on sleep quality in this study population.

Although the proportion of HCWs experiencing clinically significant mental health issues was relatively low in this cohort at the time of recruitment, prolonged work stress, isolation, and lockdowns might cause further deterioration in mental health over time. It is therefore important for governments and healthcare services to consider ways of supporting HCWs and preventing further declines in mental health. A recent systematic review on interventions shown to be effective for dealing with mental health issues of HCWs during infectious disease outbreaks reported that interventions could be grouped into four categories: (1) informational support (training, guidelines, prevention programs); (2) instrumental support (personal protective equipment, protection protocols); (3) organisational support (manpower allocation, working hours, re-organization of facilities/structures, provision of rest areas); and (4) emotional and psychological support (psychoeducation and training, mental health support team, peer-support and counselling, therapy, digital platforms and tele-support) [80]. Based on these findings, one explanation for the low prevalence of fear and distress in this cohort may relate to the high degree of informational and instrumental support provided by Loddon Mallee healthcare service leaders during the early stages of the pandemic. This is evidenced by the high proportion of HCWs who reported that their workplace was very or moderately prepared for the COVID-19 pandemic, and that they often or very often had access to the right amount of PPE. Most HCWs also had confidence in the PPE training they received and were moderately or very confident in the infection control training that they received and felt that their concerns about PPE were supported by their workplace.

This study has several strengths and some limitations. Strengths include the prospective cohort study design which will provide a detailed longitudinal understanding of the impact of the COVID-19 pandemic on the health and wellbeing of clinical and non-clinical healthcare workers in regional and rural Victoria, Australia, which to date has been largely unexplored. This will allow changes in mental health and wellbeing over time to be measured, defining the temporal sequence of events, and providing stronger evidence for a causal association with risk factors. Validated and reliable survey instruments were used for the majority of survey items, and these were selected based on robust evidence available from recent systematic reviews. Finally, response bias was evaluated by weighting study findings using workforce data which were available for approximately 80% of the study population, and the impact of this was minimal for most study measures. The main limitations of this research are that only healthy employed healthcare workers were included in this study; thus study findings may be impacted by healthy worker effects. Missing data may have introduced some sampling bias; however, the comparison of the total sample with the part of the sample weighted for non-response did not identify major differences in prevalence estimates. There were marginal differences observed for some variables. That is, the study sample had marginally fewer female, University educated, and married healthcare workers than the weighted sample, which is representative of the Loddon Mallee region sampling frame. The sample also had marginally fewer healthcare workers in part-time employment, medicine and nursing workers, healthcare workers engaging in moderate risk alcohol consumption, and workers experiencing poor-quality sleep than the sampling frame. It is also well known that female healthcare workers were more likely to be exposed to the COVID-19 virus and experienced greater levels of psychological distress during the COVID-19 pandemic compared with their male counterparts [81]. However, we did not undertake gender subgroup analyses as there was insufficient statistical power to examine differences across the large number of outcomes reported in this study due to the relatively small proportion of male healthcare workers. Finally, the sample’s representativeness may be limited to the Loddon Mallee region of regional and rural Victoria, Australia.

## 5. Conclusions

Regional and rural healthcare workers in Victoria, Australia, were experiencing a moderate to high degree of psychological distress and COVID-19-related fear during the early stages of the pandemic; however, the proportion of workers experiencing moderate to high degrees of distress was generally lower than that reported by other Australian studies and international studies. The high proportion of workers with a normal/high level of resilience, as well as the high perception of workplace pandemic preparedness and managerial support offer plausible explanations for the observed findings. Future research should examine the effects of resilience and workplace pandemic preparedness on mental health outcomes in healthcare workers, especially in the context of infectious disease outbreaks or a pandemic. The collection of the same health measures at several time-points during the first two years of the pandemic will allow us to determine whether mental health outcomes remained the same or deteriorated as the pandemic evolved and the determinants of these trends. Findings will be used to inform policy options to support healthcare workers in responding to future pandemics.

## Figures and Tables

**Figure 1 ijerph-21-00649-f001:**
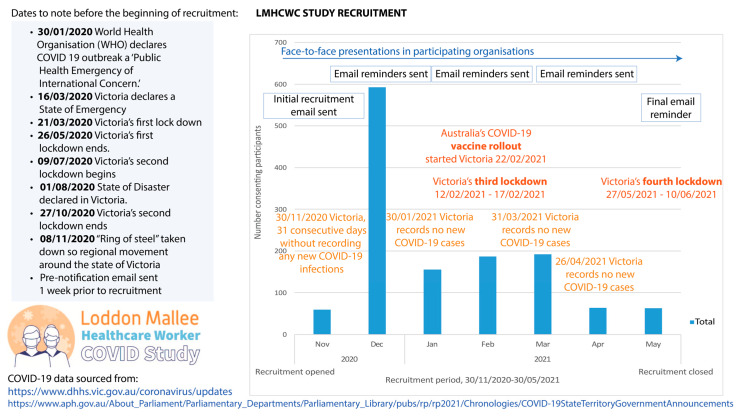
Timeline of recruitment and corresponding external COVID-19-related events for the Loddon Mallee Healthcare Worker COVID-19 Study.

**Figure 2 ijerph-21-00649-f002:**
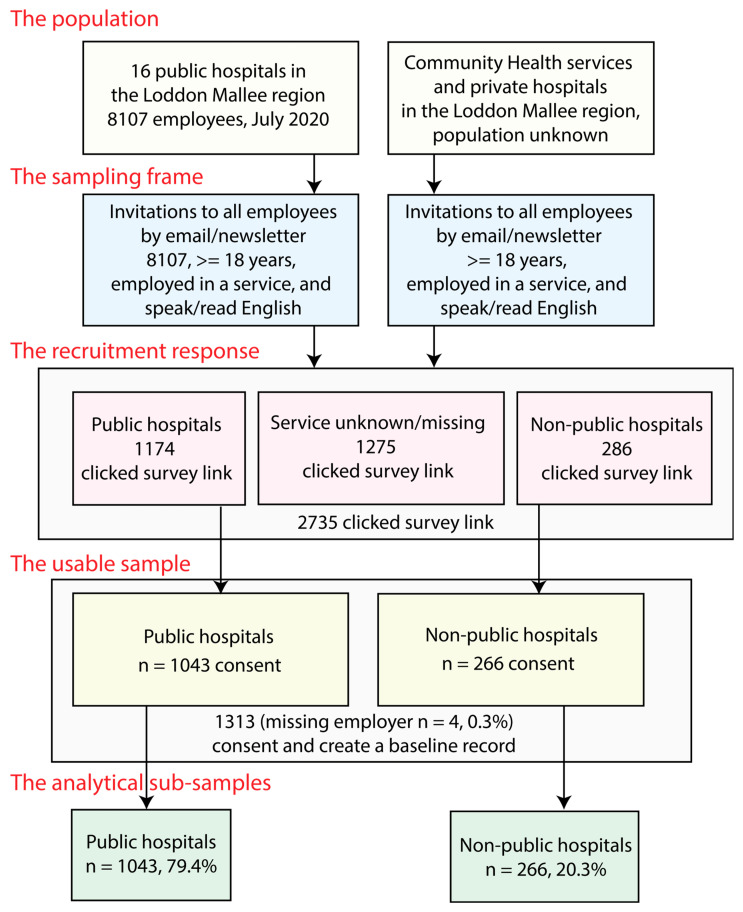
Loddon Mallee Healthcare Worker COVID-19 Study recruitment strategy and study sample.

**Table 1 ijerph-21-00649-t001:** Sociodemographic characteristics of Loddon Mallee Healthcare Worker COVID-19 Study sample recruited during the COVID-19 pandemic.

Variables	Unit of Measurement	Public Health Services				Private Health and Community Health Services		Total	
		Sample (Unweighted)		Sample Weighted for Non-Response ^#^		Sample (Unweighted)		Sample (Unweighted) *	
		N = 1043		N = 6528		N = 266		N = 1313	
Age									
	Mean (SD) (years)	44.7 (12.8)		44.3 (13.1)		45.6 (12.9)		44.9 (12.8)	
	Median (IQR)	46.0 (21.0)		45.0 (22.0)		46.0 (22.0)		46.0 (21.0)	
	Missing	31		15		9		40	
		Freq.	%	Freq.	%	Freq.	%	Freq.	%
Sex at birth									
	Female	846	81.1	5477	83.9	208	78.2	1057	80.5
	Male	140	13.4	1051	16.1	35	13.2	176	13.4
	Missing	57	5.5	0	0	23	8.6	80	6.1
Geographic location, MMM classification									
	MM 1	15	1.4	114	1.7	10	2.0	26	2.0
	MM 2	492	47.2	2864	43.9	73	43.1	566	43.1
	MM 3	68	6.5	474	7.3	91	12.2	160	12.2
	MM 4	183	17.5	1211	18.6	24	15.8	208	15.8
	MM 5	185	17.7	1282	19.6	41	17.2	226	17.2
	MM 6	3	0.3	16	0.3	0	0	3	0.2
	MM 7	1	0.1	7	0.1	0	0	1	0.1
	Missing	96	9.2	558	8.6	27	10.2	123	9.4
Household income									
	<AUD 52,000/year	163	15.6	1002	15.4	44	16.5	209	15.9
	≥AUD 52,000/year	772	69.2	4796	73.5	167	62.8	891	67.9
	Missing	158	15.1	729	11.2	55	20.7	213	16.2
Concerns about income									
	Yes	222	21.3	1431	21.9	60	22.6	284	21.6
	No	765	73.3	5070	77.7	183	68.8	950	72.4
	Missing	56	5.4	27	0.4	23	8.6	79	6.0
Education									
	Secondary Schooling Not Completed	25	2.4	148	2.3	6	2.3	32	2.4
	Secondary Schooling Completed	55	5.3	316	4.8	16	6.0	71	5.4
	Trade Qualification or TAFE	130	12.5	843	12.9	27	10.2	157	12.0
	University or Other Tertiary Study	761	73.0	5070	77.7	193	72.6	957	72.9
	Missing	72	6.9	152	2.3	24	9.0	96	7.3
Marital status									
	Married	542	52.0	3457	53.0	124	46.6	667	50.8
	De Facto/Living with Partner	173	16.6	1165	17.8	44	16.5	217	16.5
	Widowed	21	2.0	140	2.2	6	2.3	27	2.1
	Divorced/Separated	117	11.2	828	12.7	36	13.5	156	11.9
	Never Married	120	11.5	850	13.0	30	11.3	150	11.4
	Missing	70	6.7	88	1.4	26	9.8	96	7.3
Identify as Indigenous									
	Aboriginal	6	0.6	27	0.4	2	0.8	8	0.6
	Non-Aboriginal	977	93.7	6424	98.4	240	90.2	1221	93.0
	Missing	60	5.8	77	1.2	24	9.0	84	6.4
Type of Household									
	Own House	767	73.5	4955	75.9	193	72.6	962	73.3
	Own Flat/Unit	22	2.1	139	2.1	3	1.1	25	1.9
	Rented House	124	11.9	886	13.6	33	12.4	159	12.2
	Rented Flat/Unit	35	3.4	242	3.7	5	1.9	40	3.0
	Own Mobile Home	1	0.1	0	0	0	0	1	0.1
	With Family/Relatives	34	3.3	234	3.6	4	1.5	38	2.9
	With Friends	1	0.1	10	0.1	4	1.5	5	0.4
	Other	6	0.6	52	0.8	1	0.4	7	0.5
	Missing	53	5.1	10	0.1	23	8.6	76	5.8
Home schooling									
	Yes	296	28.4	1874	28.7	65	24.4	362	27.6
	No	113	10.8	769	11.8	28	10.5	142	10.8
	Missing	634	60.8	3885	59.5	173	65.0	809	61.6
Carer responsibilities									
	Yes	306	29.3	1952	29.9	67	25.2	376	28.6
	No	678	65.0	4534	69.5	174	65.4	853	65.0
	Missing	59	5.7	42	0.6	25	9.4	84	6.4

^#^ Sample characteristics weighted for baseline non-participation is the pseudo-population sample characteristics from the public hospital sample frame had the allocated number of participants responded. * Note—four participants did not provide information on whether they work within a public or private health service, thus the public and private health service subtotals do not add up to the sample total.

**Table 2 ijerph-21-00649-t002:** Sociodemographic and occupational characteristics of Loddon Mallee Healthcare Worker COVID-19 Study sample recruited during the COVID-19 pandemic according to occupational group.

	Medicine	Nursing	Allied Health	Non-Clinical	Overall Sample ^#^
	(N = 59)	(N = 499)	(N = 219)	(N = 435)	(N = 1313)
Age (years)					
>45	23 (39.0%)	259 (51.9%)	89 (40.6%)	236 (54.3%)	650 (49.5%)
≤45	36 (61.0%)	227 (45.5%)	128 (58.4%)	181 (41.6%)	623 (47.4%)
Missing	0 (0%)	13 (2.6%)	2 (0.9%)	18 (4.1%)	40 (3.0%)
Sex at birth					
Male	20 (33.9%)	49 (9.8%)	37 (16.9%)	67 (15.4%)	176 (13.4%)
Female	38 (64.4%)	442 (88.6%)	181 (82.6%)	366 (84.1%)	1057 (80.5%)
Missing	1 (1.7%)	8 (1.6%)	1 (0.5%)	2 (0.5%)	80 (6.1%)
Geographic location, MMM classification					
MM 1–2	45 (76.3%)	238 (47.7%)	102 (46.6%)	168 (38.6%)	592 (45.1%)
MM 3–7	12 (20.3%)	215 (43.1%)	96 (43.8%)	231 (53.1%)	598 (45.5%)
Missing	2 (3.4%)	46 (9.2%)	21 (9.6%)	36 (8.3%)	123 (9.4%)
Household income					
<AUD 52000/year	2 (3.4%)	70 (14.0%)	34 (15.5%)	100 (23.0%)	209 (15.9%)
≥AUD 52000/year	54 (91.5%)	368 (73.7%)	165 (75.3%)	286 (65.7%)	891 (67.9%)
Missing	3 (5.1%)	61 (12.2%)	20 (9.1%)	49 (11.3%)	213 (16.2%)
Concern about income					
No	52 (88.1%)	386 (77.4%)	164 (74.9%)	333 (76.6%)	950 (72.4%)
Yes	6 (10.2%)	111 (22.2%)	55 (25.1%)	101 (23.2%)	284 (21.6%)
Missing	1 (1.7%)	2 (0.4%)	0 (0%)	1 (0.2%)	79 (6.0%)
Education					
Other education	0 (0%)	12 (2.4%)	4 (1.8%)	12 (2.8%)	28 (2.1%)
Secondary schooling completed	0 (0%)	8 (1.6%)	5 (2.3%)	56 (12.9%)	71 (5.4%)
Secondary schooling not completed	0 (0%)	1 (0.2%)	1 (0.5%)	29 (6.7%)	32 (2.4%)
Trade qualification or TAFE	0 (0%)	49 (9.8%)	11 (5.0%)	93 (21.4%)	157 (12.0%)
University or other tertiary study	59 (100%)	429 (86.0%)	198 (90.4%)	245 (56.3%)	957 (72.9%)
Missing	0	0	0	0	68 (5.2%)
Marital status					
Married	39 (66.1%)	255 (51.1%)	115 (52.5%)	246 (56.6%)	667 (50.8%)
De facto/living with a partner	8 (13.6%)	89 (17.8%)	45 (20.5%)	65 (14.9%)	217 (16.5%)
Widowed	0 (0%)	15 (3.0%)	1 (0.5%)	10 (2.3%)	27 (2.1%)
Divorced or separated	1 (1.7%)	78 (15.6%)	17 (7.8%)	60 (13.8%)	156 (11.9%)
Never married	11 (18.6%)	54 (10.8%)	35 (16.0%)	48 (11.0%)	150 (11.4%)
Missing	0 (0%)	8 (1.6%)	6 (2.7%)	6 (1.4%)	96 (7.3%)
Home schooling					
Yes	18 (30.5%)	150 (30.1%)	61 (27.9%)	131 (30.1%)	362 (27.6%)
No	9 (15.3%)	54 (10.8%)	23 (10.5%)	51 (11.7%)	142 (10.8%)
Missing	32 (54.2%)	295 (59.1%)	135 (61.6%)	253 (58.2%)	809 (61.6%)
Caring responsibilities					
Yes	16 (27.1%)	150 (30.1%)	71 (32.4%)	131 (30.1%)	376 (28.6%)
No	42 (71.2%)	347 (69.5%)	147 (67.1%)	299 (68.7%)	853 (65.0%)
Missing	1 (1.7%)	2 (0.4%)	1 (0.5%)	5 (1.1%)	84 (6.4%)
Confidence in infection control training					
Very confident	19 (32.2%)	181 (36.3%)	60 (27.4%)	167 (38.4%)	427 (32.5%)
Moderately confident	15 (25.4%)	183 (36.7%)	98 (44.7%)	117 (26.9%)	413 (31.5%)
Somewhat confident	10 (16.9%)	71 (14.2%)	31 (14.2%)	41 (9.4%)	153 (11.7%)
A little confident	6 (10.2%)	19 (3.8%)	8 (3.7%)	10 (2.3%)	43 (3.3%)
Not confident at all	0 (0%)	8 (1.6%)	1 (0.5%)	5 (1.1%)	14 (1.1%)
Missing	9 (15.3%)	37 (7.4%)	21 (9.6%)	95 (21.8%)	263 (20.0%)
Redeployed to a new work area					
No	54 (91.5%)	397 (79.6%)	174 (79.5%)	365 (83.9%)	990 (75.4%)
Yes	4 (6.8%)	99 (19.8%)	41 (18.7%)	66 (15.2%)	210 (16.0%)
Missing	1 (1.7%)	3 (0.6%)	4 (1.8%)	4 (0.9%)	113 (8.6%)
Direct contact with COVID-19 patients					
No	26 (44.1%)	274 (54.9%)	185 (84.5%)	385 (88.5%)	870 (66.3%)
Yes	32 (54.2%)	216 (43.3%)	33 (15.1%)	43 (9.9%)	324 (24.7%)
Missing	1 (1.7%)	9 (1.8%)	1 (0.5%)	7 (1.6%)	119 (9.1%)
Considering career change due to COVID-19					
Yes	3 (5.1%)	49 (9.8%)	21 (9.6%)	38 (8.7%)	114 (8.7%)
No	54 (91.5%)	397 (79.6%)	184 (84.0%)	361 (83.0%)	1013 (77.2%)
Unsure	0 (0%)	51 (10.2%)	14 (6.4%)	35 (8.0%)	101 (7.7%)
Missing	2 (3.4%)	2 (0.4%)	0 (0%)	1 (0.2%)	85 (6.5%)

^#^ A total of 101 study participants did not indicate the type of occupation.

**Table 3 ijerph-21-00649-t003:** Occupational characteristics of Loddon Mallee Healthcare Worker COVID-19 Study sample during the COVID-19 pandemic.

Variables	Unit of Measurement	Public Health Services				Private Health and Community Health Services		Total	
		Sample (Unweighted)		Sample Weighted for Non-Response ^#^		Sample (Unweighted)		Sample (Unweighted) *	
		N = 1043		N = 6528		N = 266		N = 1313	
		Freq.	%	Freq.	%	Freq.	%	Freq.	%
Employment status									
	Full time	342	32.8	2179	33.4	92	34.6	436	33.2
	Part time	535	51.3	3761	57.6	113	42.5	649	49.4
	Casual	58	5.6	366	5.6	13	4.9	72	5.5
	Retired	9	0.9	50	0.8	2	0.8	11	0.8
	Volunteer	21	2.0	117	1.8	11	4.1	32	2.4
	Other	8	0.8	54	0.8	4	1.5	12	0.9
	Missing	70	6.7	0	0	31	11.7	101	7.7
Primary Profession									
	Medicine	46	4.4	439	6.7	12	4.5	59	4.5
	Nursing	435	41.7	3663	56.1	64	24.1	499	38.0
	Allied Health	162	15.5	599	9.2	56	21.1	219	16.7
	Non-clinical	330	31.6	1827	28.0	103	38.7	435	33.1
	Missing	70	6.7	0	0	31	11.7	101	7.7
Workplace preparedness									
	Very prepared	312	29.9	2007	30.7	52	19.5	365	27.8
	Moderately prepared	250	24.0	1672	25.6	53	19.9	303	23.1
	Somewhat prepared	189	18.1	1302	19.9	49	18.4	240	18.3
	A little prepared	103	9.9	704	10.8	30	11.3	133	10.1
	Not prepared at all	65	6.2	400	6.1	34	12.8	100	7.6
	Unsure	63	6.0	423	6.5	20	7.5	83	6.3
	Missing	61	5.8	21	0.3	28	10.5	89	6.8
Sufficient PPE for staff									
	Never	20	1.9	126	1.9	7	2.6	27	2.1
	Rarely	47	4.5	337	5.2	17	6.4	65	5.0
	Sometimes	107	10.3	755	11.6	27	10.2	134	10.2
	Often	230	22.1	1587	24.3	64	24.1	295	22.5
	Very often	405	38.8	2788	42.7	79	29.7	485	36.9
	Not applicable to my role	163	15.6	880	13.5	42	15.8	206	15.7
	Missing	71	6.8	55	0.8	30	11.3	101	7.7
Confidence in PPE training									
	Yes	764	73.3	5301	81.2	179	67.3	945	72.0
	No	66	6.3	471	7.2	14	5.3	80	6.1
	Not applicable to my role	134	12.8	699	10.7	40	15.0	175	13.3
	Missing	79	7.6	57	0.9	33	12.4	113	8.6
Workplace supported concerns about PPE									
	Yes	761	73.0	5261	80.6	178	66.9	941	71.7
	No	76	7.3	535	8.2	18	6.8	95	7.2
	Not applicable to my role	128	12.3	674	10.3	35	13.2	164	12.5
	Missing	78	7.5	58	0.9	35	13.2	113	8.6
Policy for breaks while working in full PPE									
	Yes	354	33.9	2380	36.5	79	29.7	434	33.1
	No	61	5.8	483	7.4	30	11.3	93	7.1
	Unsure	553	53.0	3637	55.7	123	46.2	677	51.6
	Missing	75	7.2	28	0.4	34	12.8	109	8.3
Redeployed to a new area of work									
	Yes	168	16.1	1143	17.5	41	15.4	210	16.0
	No	797	76.4	5340	81.8	190	71.4	990	75.4
	Missing	78	7.5	44	0.7	35	13.2	113	8.6
Confidence in new area of work (those redeployed)									
	Not confident at all	5	0.5	29	0.4	0	0	5	0.4
	A little confident	19	1.8	132	2.0	5	1.9	24	1.8
	Someday confident	40	3.8	301	4.6	11	4.1	51	3.9
	Confident	63	6.0	406	6.2	16	6.0	80	6.1
	Very confident	41	3.9	275	4.2	9	3.4	50	3.8
	Missing	875	83.9	5384	82.5	225	84.6	1103	84.0
Direct care of COVID-19 patients									
	Yes	130	12.5	906	13.9	23	8.6	153	11.7
	No	604	57.9	4183	64.1	139	52.3	747	56.9
	Not applicable to my role	233	22.3	1401	21.5	72	27.1	305	23.2
	Missing	76	7.3	38	0.6	32	12.0	108	8.2
Confidence in infection control training									
	Not confident at all	10	1.0	69	1.1	4	1.5	14	1.1
	A little confident	38	3.6	291	4.5	5	1.9	43	3.3
	Somewhat confident	128	12.3	893	13.7	25	9.4	153	11.7
	Moderately confident	333	31.9	2209	33.8	80	30.1	413	31.3
	Very confident	344	33.0	2354	36.1	80	30.1	427	32.4
	Missing	190	18.2	712	10.9	72	27.1	263	20.0
Considering career change due to COVID-19									
	Yes	83	8.0	545	8.4	30	11.3	114	8.7
	No	819	78.5	5402	82.8	191	71.8	1013	77.2
	Unsure	81	7.8	559	8.6	20	7.5	101	7.7
	Missing	60	5.8	22	0.3	25	9.4	85	6.5

^#^ Sample characteristics weighted for baseline non-participation is the pseudo-population sample characteristics from the public hospital sample frame had the allocated number of participants responded. * Note—four participants did not provide information on whether they work within a public or private health service, thus the public and private health service subtotals do not add to the sample total.

**Table 4 ijerph-21-00649-t004:** Emotional health/wellbeing of Loddon Mallee Healthcare Worker COVID-19 Study sample during the COVID-19 pandemic.

Variables	Unit of Measurement	Public Health Services				Private Health and Community Health Services		Total	
		Sample (Unweighted)		Sample Weighted for Non-Response ^#^		Sample (Unweighted)		Sample (Unweighted) *	
		N = 1043		N = 6528		N = 266		N = 1313	
Wellbeing (life satisfaction) (PWI-A) (α ^¶^ = 1.0 (single question))									
	Mean (SD)	7.2 (1.8)		7.2 (1.7)		7.2 (1.8)		7.2 (1.8)	
	Median (IQR)	8.0 (2.0)		7.3 (2.0)		8.0 (2.0)		8.0 (2.0)	
	Missing	172		1113		45		217	
Burnout (client/patient-based) (α = 0.89, 0.87–0.90)									
	Mean (SD)	24.8 (19.4)		24.8 (19.4)		24.8 (20.7)		24.9 (19.7)	
	Median (IQR)	25.0 (29.2)		25.0 (29.2)		20.8 (29.2)		25.0 (29.2)	
	Missing	337		1571		103		441	
Burnout (work/personal) (α = 0.91, 0.89–0.92)									
	Mean (SD)	46.3 (19.2)		46.6 (18.8)		44.6 (17.6)		45.9 (18.8)	
	Median (IQR)	46.4 (28.6)		46.4 (28.6)		42.9 (32.1)		46.4 (28.6)	
	Missing	805		5161		197		1005	
Isolation and loneliness (UCLA Loneliness Scale Version 3) (α = 0.87, 0.85–0.88)									
	Mean (SD)	5.0 (1.9)		5.0 (4.9)		4.8 (1.8)		4.9 (1.9)	
	Median (IQR)	5.0 (3.0)		5.0 (3.0)		4.0 (3)		5.0 (3.0)	
	Missing	224		1086		57		282	
Fear of COVID-19 scale (α = 0.89, 0.87–0.90)									
	Mean (SD)	13.05 (5.1)		12.9 (5.1)		13.26 (4.9)		13.1 (5.1)	
	Median (IQR)	13.0 (7.0)		13.0 (7.0)		13.0 (7.0)		13.0 (7.0)	
	Missing	253 (24.3%)		1247 (19.1%)		64 (24.1%)		319 (24.3%)	
Depressive symptoms (PHQ-9) (α = 0.99, 0.99–0.99)									
	Mean (SD)	5.8 (5.5)		5.7 (5.5)		5.8 (5.5)		5.9 (5.6)	
	Median (IQR)	4.0 (6.0)		4.0 (6.0)		5.0 (6.0)		4.0 (6.0)	
		Freq	%	Freq	%	Freq	%	Freq	%
	Moderate–severe								
	score ≥ 10	172	16.5	1154	17.7	36	13.5	211	16.1
	Negligible–low score < 10	663	63.6	4438	68.0	172	64.7	835	63.6
	Missing	208	19.9	936	14.3	58	21.8	267	20.3
Anxiety symptoms (GAD-7) (α = 0.93, 0.92–0.94)									
	Mean (SD)	5.5 (5.1)		5.4 (5.0)		5.5 (5.1)		5.6 (5.1)	
	Median (IQR)	5.0 (7.0)		5.0 (6.0)		4.0 (6.0)		5.0 (7.0)	
		Freq	%	Freq	%	Freq	%	Freq	%
	Moderate–severe								
	Score ≥ 10	155	14.9	1021	15.6	36	13.5	193	14.7
	Negligible–low score < 10	694	66.5	4668	71.5	182	68.4	878	66.9
	Missing	194	18.6	838	12.8	48	18.0	242	18.4
Resilience (BRS) (α = 0.90, 0.89–0.91)									
	Mean (SD)	3.5 (0.8)		3.5 (0.8)		3.6 (0.8)		3.5 (0.8)	
	Median (IQR)	3.7 (1.0)		3.7 (1.0)		3.7 (1.0)		3.7 (1.0)	
		Freq	%	Freq	%	Freq	%	Freq	%
	Low (1.00–2.99)	169	16.2	1131	17.3	38	14.3	210	16.0
	Normal (3.00–4.30)	561	53.8	3746	57.4	135	50.8	696	53.0
	High (4.31–5.00)	115	11.0	775	11.9	43	16.2	158	12.0
	Missing	198	19.0	875	13.4	50	18.8	249	19.0
Optimism (LOT-R) (α = 0.73, 0.69–0.76)									
	Mean (SD)	12.4 (2.5)		12.5 (2.5)		12.2 (2.1)		12.4 (2.4)	
	Median (IQR)	12.0 (3.0)		12.0 (3.0)		12.0 (2.0)		12.0 (3.0)	
		Freq	%	Freq	%	Freq	%	Freq	%
	Low optimism (<14)	609	58.4	4028	61.7	170	63.9	781	59.5
	Moderate optimism (14–18)	204	19.6	1392	21.3	41	15.4	246	18.7
	High optimism (>18)	16	1.5	112	1.7	2	0.8	18	1.4
	Missing	214	20.5	996	15.3	53	19.9	268	20.4
Post-traumatic stress (IES-6) (α = 0.94, 0.93–0.95)									
		Freq	%	Freq	%	Freq	%	Freq	%
	None to minimal PTS (≤9)	656	62.9	4343	66.5	173	65.0	830	63.2
	Moderate to severe PTS (>9)	181	17.4	1226	18.8	46	17.3	230	17.5
	Missing	206	19.8	959	14.7	47	17.7	253	19.3
Fatigue									
		Freq	%	Freq	%	Freq	%	Freq	%
	All the time	22	2.1	142	2.2	5	1.9	27	2.1
	Most of the time	202	19.4	1401	21.5	45	16.9	248	18.9
	A good bit of the time	208	19.9	1390	21.3	60	22.6	268	20.4
	Some of the time	265	25.4	1732	26.5	57	21.4	322	24.5
	A little of the time	142	13.6	953	14.6	38	14.3	182	13.7
	None of the time	34	3.3	221	3.4	14	5.3	49	3.7
	Missing	170	16.3	689	10.5	47	17.7	217	16.5

^#^ Sample characteristics weighted for baseline non-participation is the pseudo-population sample characteristics from the public hospital sample frame had the allocated number of participants responded. * Note—four participants did not provide information on whether they work within the public or private health service, thus the public and private health service subtotals do not add to the sample total. ^¶^ α = Cronbach’s alpha and 95% confidence interval (internal consistency)).

**Table 5 ijerph-21-00649-t005:** Emotional health and wellbeing of Loddon Mallee Healthcare Worker COVID-19 Study sample recruited during the COVID-19 pandemic according to occupational group.

Characteristic	Medicine N = 59	Nursing N = 499	Allied Health N = 219	Non-Clinical N = 435	*p*-Value ^$^	N = 1313 ^#^
Wellbeing (life satisfaction) (PWI-A Queston 1)					0.1	
Mean (SD)	7.7 (1.4)	7.2 (1.9)	7.2 (1.7)	7.2 (1.8)		7.2 (1.8)
Median (IQR)	8.0 (1.5)	8.0 (2.0)	8.0 (2.0)	8.0 (2.0)		8.0 (2.0)
Missing	8	60	16	32		217
Burnout (client/patient-based)					<0.001	
Mean (SD)	31.4 (20.3)	24.7 (19.4)	27.3 (19.8)	21.1 (19.4)		24.9 (19.7)
Median (IQR)	29.2 (26.6)	25.0 (29.2)	25.0 (25.0)	16.7 (29.1)		25.0 (29.2)
Missing	4	66	31	239		441
Burnout (work/personal)					0.084	
Mean (SD)	60.7 (16.4)	48.8 (17.5)	51.4 (22.1)	44.5 (18.6)		45.9 (18.8)
Median (IQR)	64.3 (16.0)	50.0 (28.6)	53.6 (25.0)	42.9 (28.5)		46.4 (28.6)
Missing	56	450	194	204		1005
Isolation and loneliness (UCLA Loneliness Scale Version 3)					0.61	
Mean (SD)	5.0 (1.9)	5.0 (1.9)	4.9 (1.8)	5.0 (1.9)		4.9 (1.9)
Median (IQR)	5.0 (3.0)	5.0 (3.0)	4.0 (3.0)	5.0 (3.0)		5.0 (3.0)
Missing	13	95	24	49		282
Fear of COVID-19 Scale					0.039	
Mean (SD)	11.2 (4.0)	13.3 (5.1)	13.6 (5.3)	12.9 (5.1)		13.1 (5.1)
Median (IQR)	10.0 (6.0)	13.0 (7.0)	13.0 (8.0)	12.0 (7.0)		13.0 (7.0)
Missing	15	103	33	67		319
Depressive symptoms (PHQ-9)					0.66	
Mean (SD)	5.3 (5.6)	5.8 (5.7)	6.0 (5.7)	5.9 (5.5)		5.9 (5.6)
Median (IQR)	4.0 (5.0)	4.0 (6.0)	4.0 (7.0)	5.0 (6.0)		4.0 (6.0)
Missing	8	78	26	54		267
Depressive symptoms group (PHQ-9)					0.78	
Negligible–low score < 10	43 (73.0%)	331 (63.3%)	155 (70.8%)	306 (70.3%)		835 (63.6%)
Moderate–severe ≥ 10	8 (13.5%)	90 (18.0%)	38 (17.3%)	75 (17.2%)		211 (16.1%)
Missing	8	78	26	54		267
Anxiety symptoms (GAD-7)					0.39	
Mean (SD)	4.6 (5.1)	5.6 (5.05)	5.7 (5.3)	5.5 (5.1)		5.6 (5.1)
Median (IQR)	3.0 (5.0)	5.0 (7.0)	5.0 (5.0)	5.0 (7.0)		5.0 (7.0)
Missing	10	67	22	42		242
Anxiety symptoms group					0.94	
Negligible–low score < 10	41 (69.5%)	352 (70.5%)	164 (74.9%)	321 (73.8%)		878 (66.9%)
Moderate–severe ≥ 10	8 (13.6%)	80 (16.0%)	33 (15.1%)	72 (16.5%)		193 (14.7%)
Missing	10	67	22	42		242
Resilience (BRS)					0.13	
Mean (SD)	3.8 (0.82)	3.5 (0.8)	3.5 (0.8)	3.5 (0.8)		3.5 (0.8)
Median (IQR)	3.8 (1.0)	3.7 (1.0)	3.7 (1.0)	3.7 (1.0)		3.7 (1.0)
Missing	10	74	23	41		249
Resilience group					0.069	
Low (1.00–2.99)	5 (8.5%)	86 (17.2%)	42 (19.2%)	77 (17.7%)		210 (16.0%)
Normal (3.00–4.30)	29 (49.1%)	280 (56.1%)	128 (58.5%)	259 (59.5%)		696 (53.0%)
High (4.31–5.00)	15 (25.4%)	59 (11.8%)	26 (11.9%)	58 (13.3%)		158 (12.0%)
Missing	10	74	23	41		249
Optimism (LOT-R)					0.002	
Mean (SD)	11.5 (2.2)	12.5 (2.4)	11.9 (2.0)	12.6 (2.6)		12.4 (2.4)
Median (IQR)	12.0 (3.0)	12.0 (3.0)	12.0 (2.0)	12.0 (3.0)		12.0 (3.0)
Missing	13	84	20	50		268
Optimism group						
Low optimism (<14)	40 (67.8%)	304 (60.9%)	160 (73.1%)	277 (63.7%)		781 (59.5%)
Moderate optimism (14–18)	6 (10.2%)	103 (20.6%)	38 (17.3%)	99 (22.8%)		246 (18.7%)
High optimism (>18)	0 (0%)	8 (1.6%)	1 (0.5%)	9 (3.8%)		18 (1.4%)
Missing	13	84	20	50		268
Post-traumatic stress (IES-6)					0.71	
Mean (SD)	4.8 (5.1)	5.0 (5.2)	4.6 (5.4)	5.2 (5.9)		5.0 (5.5)
Median (IQR)	4.0 (8.0)	3.0 (8.0)	3.0 (6.0)	3.0 (8.0)		3.0 (8.0)
Missing	9	78	17	48		253
Post-traumatic stress group (IES-6)					0.14	
None to minimal PTS (≤9)	39 (66.1%)	327 (65.5%)	170 (77.6%)	294 (67.6%)		830 (63.2%)
Moderate to severe PTS (>9)	11 (18.6%)	94 (18.8%)	32 (14.6%)	93 (21.4%)		230 (17.5%)
Missing	9	78	17	48		253
Fatigue					0.86	
All of the time	1 (2.0%)	8 (1.8%)	6 (2.9%)	12 (3.0%)		27 (2.5%)
Most of the time	14 (27%)	96 (22%)	39 (19%)	99 (25%)		248 (23%)
A good bit of the time	11 (22%)	113 (26%)	55 (27%)	89 (22%)		268 (24%)
Some of the time	14 (27%)	128 (29%)	60 (29%)	120 (30%)		322 (29%)
A little of the time	9 (18%)	75 (17%)	35 (17%)	63 (16%)		182 (17%)
None of the time	2 (3.9%)	17 (3.9%)	10 (4.9%)	20 (5.0%)		49 (4.5%)
Missing	8	62	14	32		217

^#^ A total of 101 study participants did not indicate the type of occupation. ^$^ Kruskal–Wallis rank sum test; Pearson’s chi-squared test.

**Table 6 ijerph-21-00649-t006:** Comparison * of continuous and binary outcomes for measures of emotional health and wellbeing across occupations.

Variables	Medicine vs. Nursing Mean Difference	Allied vs. Nursing Mean Difference	Non-Clinical/ Administration vs. Nursing Mean Difference	Medicine vs. Allied ^#^ Mean Difference	*p*-Value	Medicine vs. Nursing Difference	Allied vs. Nursing Difference	Non-Clinical/ Administration vs. Nursing Difference	Medicine vs. Allied ^#^ Difference	*p*-Value **
Comparison of Continuous Outcomes						Comparison of Binary Outcomes				
Wellbeing (life satisfaction (PWI-A—Q1)	0.7 (0.1, 1.3)	0.2 (−0.2, 0.5)	0.1 (−0.2, 0.5)	0.5 (−0.1, 1.2)	0.037	NA	NA	NA	NA	NA
Burnout (client/patient-based)	5.0 (−2.6, 12.6)	2.2 (−2.5, 6.9)	−1.7 (−6.9, 3.5)	2.8 (−5.3, 10.9)	0.144	NA	NA	NA	NA	NA
Burnout (work/personal)	17.8 (−17.3, 52.9)	−2.8 (−15.9, 10.3)	−6.9 (−15.8, 1.9)	20.6 (−15.2, 56.4)	0.055	NA	NA	NA	NA	NA
Isolation and loneliness (UCLA Loneliness Scale Version 3)	0.1 (−0.7, 0.9)	−0.2 (−0.6, 0.2)	0.01 (−0.4, 0.4)	0.3 (−0.5, 1.1)	0.611	−0.05 (−0.9, 0.8)	0.3 (−0.2, 0.8)	−0.1 (−0.5, 0.3)	−0.3 (−1.2, 0.6)	0.345
Fear of COVID−19 Scale	−1.6 (−3.7, 0.6)	0.5 (−0.7,1.7)	−0.5 (−1.5, 0.6)	−2.1 (−4.4, 0.1)	0.047	NA	NA	NA	NA	NA
Depressive symptoms (PHQ−9)	−1.2 (−3.4, 1.0)	−0.2 (−1.5, 1.1)	0.01 (−1.1, 1.1)	−0.97 (−3.3, 1.3)	0.5448	−0.8 (−2.0, 0.4)	−0.4 (−1.0, 0.2)	−0.2 (−0.8, 0.3)	−0.4 (−1.7, 0.8)	0.165
Anxiety symptoms (GAD−7)	−1.4 (−3.5, 0.6)	−0.2 (−1.4, 1.0)	−0.1 (−1.1, 0.9)	−1.2 (−3.4, 0.9)	0.355	−0.4 (−1.6, 0.8)	−0.4 (−1.0, 0.3)	−0.1 (−0.7, 0.4)	−0.04 (−1.3, 1.2)	0.468
Post-traumatic stress (IES−6)	−0.4 (−2.5, 1.8)	−0.6 (−1.9, 0.6)	0.2 (−0.9, 1.3)	0.3 (−2.0, 2.6)	0.457	−0.2 (−1.2, 0.8)	−0.6 (−1.2, 0.03)	−0.0 (−0.5, 0.5)	0.4 (−0.7, 1.5)	0.084
Resilience (BRS)	0.3 (−0.03, 0.6)	−0.05 (−0.2, 0.1)	0.04 (−0.1, 0.2)	0.3 (−0.0, 0.7)	0.069	1.0 (−0.4, 2.4)	0.01 (−0.6, 0.6)	0.14 (−0.4, 0.7)	0.97 (−0.5, 2.4)	0.313
Optimism (LOT-R)	−0.9 (−1.9, 0.1)	−0.7 (−1.2, −0.1)	−0.03 (−0.5, 0.5)	−0.2 (−1.3, 0.8)	0.004	0.9 (−0.3, 2.2)	0.5 (−0.1, 1.1)	0.0 (−0.5, 0.5)	0.4 (−0.9, 1.8)	0.048

^#^ With Tukey’s adjustment. The binary differences are given on the log odds ratio (not the response) scale. * The models used to estimate differences between means or relative risks are adjusted for age, gender, modified Monash category, and education. Relative risks correspond to the risk of having the poorer psychological outcome category. ** *p*-values correspond to the test of the global null hypothesis of no difference between occupations.

**Table 7 ijerph-21-00649-t007:** Lifestyle characteristics of Loddon Mallee Healthcare Worker COVID-19 Study sample during the COVID-19 pandemic.

Variables	Unit of Measurement	Public Health Services				Private Health and community Health Services		Total	
		Sample (Unweighted)		Sample Weighted for Non-Response ^#^		Sample (Unweighted)		Sample (Unweighted) *	
		N = 1043		N = 6528		N = 266		N = 1313	
		Freq	%	Freq.	%	Freq.	%	Freq.	%
Body mass index (BMI) kgm^2^									
	<18.5	8	0.8	62	1.0	2	0.8	10	0.8
	18.5–24.9	158	15.1	994	15.2	44	16.5	202	15.4
	25.0–29.9	173	16.6	1137	17.4	39	14.7	213	16.2
	30.0–39.9	172	16.5	1221	18.7	47	17.7	220	16.8
	≥40	40	3.8	256	3.9	12	4.5	53	4.0
	Missing	492	47.2	2857	43.8	122	45.9	615	46.8
Alcohol intake (standard drinks/day)									
	Non-drinker/ low-risk drinker (0–4)	450	43.1	2870	44.0	123	46.2	574	43.7
	Moderate-risk drinker (5–7)	160	15.3	1137	17.4	32	12.0	192	14.6
	High-risk drinker (8–12)	60	5.8	390	6.0	18	6.8	79	6.0
	Missing	373	35.8	2132	32.7	93	35.0	468	35.6
Smoking									
	Never	504	48.3	3264	50.0	129	48.5	634	48.2
	Ex-smoker	205	19.7	1407	21.6	49	18.4	254	19.3
	Current	69	6.6	487	7.5	25	9.4	96	7.3
	Missing	265	25.4	1370	21.0	63	23.7	329	25.0
Dietary intake No. serves of vegetables/day									
	1 serve or less	94	9.0	639	9.8	24	9.0	118	9.0
	2–3 serves	398	38.2	2658	40.7	110	41.4	510	38.8
	4–5 serves	181	17.4	1148	17.6	51	19.2	232	17.7
	6 serves or more	40	3.8	267	4.1	6	2.3	46	3.5
	Do not eat vegetables	5	0.5	45	0.7			5	0.4
	Missing	325	68.8	1771	27.1	75	28.2	402	30.6
Dietary intake No. serves of fruit/day									
	1 serve or less	312	29.9	2057	31.5	79	29.7	391	29.8
	2–3 serves	344	33.0	2295	35.2	94	35.3	440	33.5
	4–5 serves	34	3.3	224	3.4	12	4.5	46	3.5
	6 serves or more	8	0.8	66	1.0	2	0.8	10	0.8
	Do not eat fruit	19	1.8	119	1.8	4	1.5	23	1.8
	Missing	326	31.3	1767	27.1	75	28.2	403	30.7
Sleep (PSQI) (α ^¶^ = 0.81, 0.74–0.85)									
	Poor sleep quality	516	49.5	3429	52.5	113	42.5	631	48.0
	Good sleep quality	201	19.3	1337	20.5	73	27.4	274	20.9
	Missing	325	31.2	1750	26.8	80	30.1	407	31.0

^#^ Sample characteristics weighted for baseline non-participation is the pseudo-population sample characteristics from the public hospital sample frame had the allocated number of participants responded. * Note—four participants did not provide information on whether they work within a public or private health service, thus the public and private health service subtotals do not add to the sample total. ^¶^ α = Cronbach’s alpha and 95% confidence interval (internal consistency)).

## Data Availability

Due to privacy reasons, subsets of non-identifiable data may be made available to collaborating researchers where there is a formal request (i.e., expression of interest research proposal) to make use of the data. Collaborators may request to either analyse LMHCWCS questionnaire data and/or conduct a substudy of the LMHCWCS participants. For both, permission must be first obtained from the Data Access Committee of LMHCWCS, or in the case of substudies, the Investigator Committee. All future collaborators will be linked to the project through a Human Research Ethics Committee amendment that outlines who will be using the data and the purpose of the project [4]. Shared data cannot be shared with any third party and may only be used for the purposes approved by the study’s Data Access Committee. Researchers wishing to access data should contact the Principal Investigator using the corresponding author contact details or the Study Coordinator at loddonmalleeHCWCOVIDstudy@lmss.org.au.

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
