# Peer review of "Health and Wellbeing of Regional and Rural Australian Healthcare Workers during the COVID-19 Pandemic: Baseline Cross-Sectional Findings from the Loddon Mallee Healthcare Worker COVID-19 Study—A Prospective Cohort Study"

_ijerph, 2024, doi:10.3390/ijerph21050649_

Round 1

Reviewer 1 Report

Comments and Suggestions for Authors

This study focus on the immediate and long-term impacts of the COVID-19 pandemic on regional and rural healthcare workers. Authors found that regional/rural healthcare workers were experiencing a moderate to high degree of psychological distress during the early stages of the pandemic. However, most participants demonstrated a normal/high degree of resilience. I have just some minor comments.

1. Methods, the authors were advised to give the Cronbachs alpha coefficient of the original scale and the Cronbachs alpha coefficient in the present study.

2. Results, Table 6 using a horizontal format may be more beneficial to reading.

3. Discussion, the discussion needs to go beyond just stating consistency with previous research and expanding on possible causes, etc., and overall, it is not deep enough.

4. Discussion, the authors may consider adding a section related to recommendations for authorities and managers or even the public health profession as a whole based on the results of this study as appropriate.

5. Conclusions, the authors may consider streamlining this section to allow for more focus.

6. Tables should be produced in accordance with the norms of textual publication, e.g. a three-line table should be used.

7. Overall, I believe the manuscript may benefit from a thorough review of the language.

Comments on the Quality of English Language

Minor editing of English language required.

Author Response

Best wishes,

Mark McEvoy

Reviewer 2 Report

Comments and Suggestions for Authors

Thank you for the opportunity to review this manuscript. The authors present a cross-sectional analysis of baseline data from 1313 healthcare workers in regional and rural areas within Victoria, Australia, who have been recruited for a prospective cohort study. However, the title and abstract could be clearer in stating that this is a cross-sectional analysis. Validated measures were used to determine symptoms of depression, anxiety, post-traumatic stress, burnout, resilience, optimism, and fear of COVID-19. I have provided some minor points below:

-          In the title and abstract, please make it clearer that this paper refers only to cross-sectional analyses of baseline data.

-          The literature review provides a thorough overview of existing literature and demonstrates the need for this research.

-          The aims are more relevant to the overall aims of the prospective cohort study, rather than the present cross-sectional analyses. For example, it is not possible from the analysis presented to prospectively describe the health and wellbeing impacts or to identify factors which predict better physical and mental health outcomes. Please consider revising the aims to reflect the aims of this cross-sectional analysis.

-          The methods are described in sufficient detail. Validated measures and criteria were used to record outcomes.

-          The figures presented in the total column in Table 4 do not match up with the figures presented in the total column in Table 5. The percentages for depression, anxiety, resilience, optimism, and PTSD are very different. It is not clear why this is the case.

-          There are many tables and some of the results are repeated. Could the authors consider moving some results tables into supplementary materials?

Author Response

Best wishes,

Mark McEvoy

Round 2

Reviewer 1 Report

Comments and Suggestions for Authors

No suggestions, agree to publish.